# Sexual Harassment and Assault across Trail and Ultrarunning Communities: A Mixed-Method Study Exploring Gender Differences

Christy Teranishi Martinez *, Crista Scott Tappan, Harley Baker, Makayla Edwards and Juliane Martinez

Department of Psychology, California State University Channel Islands, Camarillo, CA 93012, USA; cristatappan@gmail.com (C.S.T.); harley.baker@csuci.edu (H.B.); makayla.edwards54@gmail.com (M.E.); martinezjulie884@gmail.com (J.M.)
* Correspondence: christy.teranishi-martinez@csuci.edu

**Abstract:** This mixed-method study aimed to better understand the prevalence and qualitative experiences of sexual harassment and assault (SHSA) among trail and ultrarunners. Over 1500 runners (1215 females; 259 males; 28 transgender, non-binary, gender-fluid) of ages ranging from 18 to 77 (*M* = 39) responded to an online survey assessing the frequency and types of SHSA incidents experienced and the extent to which SHSA changed running behavior and feelings of safety. Respondents reported between 0 and over 100,000 incidents of SHSA, including catcalls, spanking, flashing, unwanted verbal advances, stalking, forced sexual acts, and rape. Of the 1502 respondents, 61% reported SHSA while running. Significantly higher rates of SHSA were reported by female, transgender, non-binary, and gender-fluid runners compared to male runners: 70% of female respondents and 61% of transgender, non-binary, and gender-fluid respondents reported incidents of SHSA compared to 17% of male respondents (all $p < 0.001$). Utilizing Mahalanobis procedures, discriminant, and chi-square analyses, a group of 75 responders was identified as outliers, reporting significantly more incidents of SHSA (1000 to 300,000) than the more normative respondents ($p < 0.0001$). For each type of SHSA, the two groups differed significantly in the number of reported incidents ($p < 0.001$). There were significant differences in perceived safety and how SHSA changed their running behavior. Gender inequities and challenging false memory claims are discussed. The findings underscore the importance of empowering runners to create a shared vision for the running community to promote safety and well-being.

**Keywords:** ultrarunners; sexual harassment; sexual assault; gender inequities; mixed method

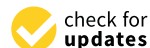



## 1. Introduction

For many, running is seen as a refuge associated with a myriad of benefits, including physical fitness, mental health, *flow* (e.g., complete absorption in an enjoyable activity), community engagement, and a sense of achievement (Basson 2001; see review by Oswald et al. 2020; Teranishi Martinez and Scott 2016). Running 50 kilometers to over 150K, trail and ultrarunning is the fastest-growing subdivision of the running community (Ronto 2021). While researchers have emphasized the benefits of running for physical, mental, and spiritual well-being, it is essential to be aware of the adverse effects and risks associated with running outdoors and on secluded trails, especially since the prevalence of sexual harassment and assault (SHSA) is on the rise (RAINN n.d.).

According to the Centers for Disease Control, approximately 75% of women and 30% of men have reported sexual harassment and 50% of women and 17% of men have reported sexual assault in their lifetime (Basile et al. 2014). Over the past 20 years, a growing body of research has indicated an increase in SHSA across a wide array of sports and athletic activities (Ayala et al. 2021; Fasting 2016; Gündüz et al. 2008; Messner 2002; Rintaugu et al. 2014). As the prevalence of SHSA across various sports has increased, so has the

number of incidents in the running community. When several U.S. women runners were victims of homicide in 2016, fear and anger sparked throughout the community, prompting Runner's World (Hamilton 2017) to survey runners' experiences and responses to SHSA. Of the 4670 respondents, 43% of women reported being harassed while running, which increased to 50% for those between 30 and 35 years old and to 58% for women under 30. Four percent of men also reported being catcalled, stalked, or sexually assaulted while running. While these statistics suggest a trend among runners, there is a lack of empirical research examining the prevalence of SHSA in the trail and ultrarunning community.

The present study aims to fill this gap while providing runners with a safe space to share their experiences of SHSA in trail and ultrarunning communities worldwide. Since existing research in the field is limited, the following literature review will examine the prevalence of SHSA among exercisers and athletes across various sports and explore gender differences. Building on previous studies, this study will investigate the impact of SHSA across trail and ultrarunning communities to provide data to stakeholders and policymakers so resources can be allocated toward more effective prevention strategies and intervention tactics, and support for victims.

### 1.1. Prevalence of SHSA in Sports

Substantial evidence shows that SHSA is prevalent across different sports and athletic activities. Mountjoy et al. (2016) found that the overall lifetime prevalence of sexual harassment in sports ranged from 19% to 92%, and the prevalence of sexual abuse ranged from 2% to 49%. Examining 41 studies on gender-based violence in sports across 17 countries, Lang et al. (2023) pointed out that the difficulty of comparing results across studies stems from the wide range of definitions, research designs, methodologies, and samples used. Sexual harassment was the most frequently used term. Although sexual violence was the second most frequently used term, it encompassed sexual harassment and sexual abuse. The terms sexual abuse, sexual harm, sexual assault, emotional/psychological violence, physical violence, and homophobic violence were also frequently used across studies. In addition, some studies only consisted of a female sample. Lang et al. emphasized the need to disaggregate data to better understand the antecedents and prevalence of SHSA across various sports and athletic activities and the demographic factors correlated with SHSA.

### 1.2. Setting and Performance Level

The news often sensationalizes isolated incidents of the SHSA of athletes who are sexually harassed or abused by coaches or teammates. To understand the prevalence of SHSA, it is essential to examine the setting in which SHSA occurs. In a cross-sectional retrospective study of 370 Australian elite-level and club athletes, Leahy et al. (2002) found that 41% of females and 29% of males reported being sexually assaulted while playing sports. In a cross-sectional study of 1214 Nigerian university male and female athletes, Elendu and Umeakuka (2011) found that 30% reported unwanted sexual attention from coaches, whereas 80% reported sexual harassment and 86% reported sexual coercion by teammates. Findings suggest that within the context of athletics and sports, SHSA are not isolated incidents but are perpetuated by a system that allows them to occur.

Researchers have examined whether athletes are more likely to experience sexual harassment compared to those not participating in sports and whether the type of sport or performance level predicts SHSA (Fasting et al. 2004; Messner 2002; Messner and Sabo 1994; Volkwein et al. 1997). In a study of 595 Czechoslovakian women, Fasting et al. (2010) examined the prevalence of SHSA by comparing three performance groups, including elite-level athletes (*n* = 229), non-elite-level athletes (*n* = 224), and exercisers (*n* = 142). Regardless of performance level, 72% of all participants reported experiencing some form of sexual harassment. However, significant differences were found within sports. Elite athletes were more likely to report SHSA (55%) than non-elite athletes (29%), indicating a possible increased risk of SHSA in elite-level sports. Outside of sports, 73% of exercisers reported SHSA. Findings indicate that SHSA is prevalent across all contexts of

physical fitness, athletics, and sports, signifying the need to find solutions to alleviate this pervasive problem.

### 1.3. Gender Disparities

Gender disparities in reported incidents of SHSA in sports are evident (Lang et al. 2023). One in six men compared to one in three women have reported experiencing sexual violence in their lifetime (Smith et al. 2018). These gender disparities are congruent with Lieu and Rennison's (2018) study within the climbing community. Of 5311 climbers, 47% of women and 16% of men reported at least one incident of SHSA while climbing. Female climbers were more likely to experience multiple types of SHSA victimization than male climbers and most frequently experienced verbal harassment (57%), catcalls (55%), unwanted touching (41%), and unwanted following (40%). Male climbers who reported SHSA most frequently experienced catcalls (49%), unwanted touching (29%), and flashing (19%). Survivors reported that SHSA changed how they participate in their sport, causing them to disengage from the climbing community, reduce travel for climbing purposes, and limit whom they climb with.

In a longitudinal study from 1997 to 1999 consisting of 553 Norwegian elite female athletes across 56 different sports disciplines, 165 females reported experiencing SHSA in a sport setting (Fasting et al. 2004). Fasting et al. (2010) found that among 616 female sports students from the Czech Republic, Greece, and Norway, 37% reported experiencing sexual harassment while playing sports. In Turkey, Gündüz et al. (2008) found that among 356 female university athletes, 200 had reported at least one or more incidents of sexual harassment. In Kenya, Rintaugu et al. (2014) found that of 339 female student-athletes surveyed, 64% reported experiencing at least one experience of sexual harassment while playing sports, while 45% reported experiencing harassment two to five times. The most common forms of sexual harassment were sexually offensive looks, comments, and unwanted comments on attractiveness in public. Studies that research the experience of SHSA in sports contribute to understanding the prevalence of SHSA and how to prevent it from occurring. Yet, studies only including female athletes perpetuate gender-based inequalities by not protecting male and other gender-diverse athletes who experience SHSA.

A significant limitation is that some of these studies do not include a representative sample or disaggregate data to examine important sub-categories, such as marginalized gender and sexual minorities. Approximately 50% of transgender, non-binary, and gender-fluid individuals report having been sexually assaulted in their lifetime, and 1 in 10 report sexual assault within the past year (James et al. 2016). Few studies include gender-diverse athletes identifying as transgender, non-binary, gender-fluid, Two-Spirit, etc., which is highly problematic, as they may be more vulnerable to SHSA while engaging in sports (Taha-Thomure et al. 2022; Vertommen et al. 2016). The inclusion of gender-diverse athletes is important for revealing inequities, exposing hidden trends, and identifying vulnerable populations to understand the scope of the problem entirely.

Sports have traditionally been encompassed by stereotypical male-dominant cultural norms, contributing to various forms of gender inequities and SHSA (Fasting et al. 2010; Krauchek and Ranson 1999). Traditional beliefs and expectations about masculinity are associated with cultural expectations for boys to initiate sex, be sexually aggressive with girls, and adhere to peer pressure, viewing sexual conquest as a validation for male adequacy (Zilbergeld 1993). Supporting beliefs and values that condone these behaviors has been found to be a strong predictor of sexual violence (Good and Wood 1995; Mosher and Anderson 1986). It is essential to examine the various factors that contribute to the system and culture of sports that allow this problem to persist.

### 1.4. Present Study

To date, no empirical studies have examined SHSA across trail and ultrarunning communities. The main objective of the present study is to understand the extent of gender disparities in the frequency and types of incidents reported by trail and ultrarunners and

how this impacts perceived safety within the running community. A second aim of this study is to create a safe space to share stories and testimonials of SHSA to empower runners to have a voice to urge stakeholders and policymakers to allocate time and resources toward alleviating the problem. This mixed-method study incorporates quantitative and qualitative analyses to better understand the prevalence and qualitative experiences of reported incidents of SHSA among trail and ultrarunners. Building on previous research, we predict that there are gender differences in the reported frequency and types of incidents of SHSA and that frequency and types of SHSA would predict perceptions of safety and changes in running behavior.

### 1.5. Definitions

As discussed earlier, various definitions, measures, and research designs make SHSA challenging to assess (Lang et al. 2023). In addition, SHSA may be construed differently based on individuals' perceptions and experiences in relation to gender, age, and frequency of victimization. For this study, the following definitions will be used regarding the number and type of incidents that individuals in the trail and ultrarunning community report while engaged in a running-related activity. *Sexual harassment* is defined as physical or verbal harassment of a sexual nature and unwelcomed sexual advances, including but not limited to catcalls, flashing, and requests for sexual favors (Kearl 2018). *Sexual assault* is non-consensual sexual contact or a sexual act someone was forced to do against their will, such as forced kissing, fondling, unwanted touching, spanking, oral sex, and rape (Kearl 2018).

## 2. Method

### 2.1. Participants

Although 92% of the participants were from the United States (*n* = 1380), participants in this study responded from 17 different countries, representing every continent across the world. Of the 1502 respondents, 17% identified as male (*n* = 259), 81% identified as female (*n* = 1215), and 2% identified as transgender, non-binary, or gender-fluid (*n* = 28). Ages ranged from 18 to 77 years old (*M* = 39.64; *SD* = 9.58). Of the 1502 participants, 86% identified as White, Caucasian, European, or European American; 5.4% as Latino/a, Chicano/a, or Hispanic; 2.4% as Asian American/Pacific Islander; 0.8% as African American or Black; 0.8% as Native American, American Indian, or Indigenous; 4.1% as mixed ethnicity/race/culture; and 0.5% identified as Other.

### 2.2. Research Design

In this mixed-method study, an online survey was developed to assess the types of SHSA incidents and to examine how SHSA changed the quantity and quality of running and perceptions of safety in the running community. A self-report questionnaire developed by Lieu and Rennison (2018) was adapted for the running community. Survey questions included: "Have you ever experienced sexual harassment or sexual assault while engaged in a running-related activity?; If yes, what types of SHSA incidents have you experienced, and how often did these incidents occur?; Have these SHSA incidents changed your relationship with running?; And if you are comfortable, please share your experience(s) or about any other thoughts on the topic."

### 2.3. Procedures

After obtaining IRB approval from our affiliated university, non-probability convenience sampling was used to administer our online survey using Qualtrics. Through word of mouth, email, social media, and websites, we recruited over 1000 responses in two weeks. Runner's World magazine published an article on the goals of this study, resulting in another 1000 responses. After removing incomplete, repetitive, obscene, and incomprehensible data, the final sample consisted of 1502 responses.

*2.4. Data Screening and Analysis*

We hypothesized gender differences in the reported frequency and types of incidents of SHSA, and that the frequency and types of SHSA would predict the perception of safety and changes in running behavior. The data were submitted to standard reliability and validity analysis (e.g., Tabachnick and Fidell 2013). This included testing distributional assumptions, the presence of extreme outliers, the probable effects of missing data, and ways to minimize those effects. We expected the SHSA frequency data, like most data based on counts or frequency, to fit either a Poisson or Negative Binomial distribution. Under these assumptions, the data would be analyzed using Poisson (Cameron and Trivedi 1998) or Negative Binomial (Hilbe 2007) regression models. Both models are widely used with count data. Unfortunately, neither the SHSA types nor the total number of reported incidents conformed to either distribution (all $p < 0.0001$). Further analysis showed the cause to be a group of 75 extreme outliers identified through Mahalanobis procedures with follow-up discriminant and chi-square analyses. These 75 respondents, representing 5% of the overall sample, were compared to the rest of the sample on their total reported number of SHSA experiences. Non-outliers reported an average of 18.53 SHSA incidents, while the outliers reported an average of 518.70 SHSA experiences. An unequal variances *t*-test of the Mahalanobis *p*-values[1] confirmed that the outliers were substantially different from the rest of the sample $t(1426.222) = 232.174$, $p < 0.00001$, $d = 6.63$. While the value of both *t* and *p* are somewhat suspect because of the large sample size, Cohen's *d* (6.63) is not, indicating that the extreme outlier group is more than six standard deviations away from the rest of the sample in terms of the number of reported SHSA incidents. As such, the sample appeared to have both "normative responders" and "extreme responders" in terms of the number of SHSA incidents they reported.

These extreme responders posed insurmountable problems and required rethinking the original analysis plan. Rather than eliminate the extreme responding participants, participant responses were dichotomized on all SHSA frequency data into two categories: (a) reporting no SHSA experiences, and (b) reporting at least one SHSA experience. This transformed the SHSA responses into "No/Yes" (0, 1) data that fit most parametric analyses and almost all nonparametric analyses. In this way, data from all the respondents were included in the analysis. Missing data were deemed nonproblematic with fewer than 2% of the respondents having missing data.

**3. Results**

Of the 1502 participants, 61% reported that they experienced sexual harassment and/or assault while running ($n = 909$), 34% did not ($n = 517$), 5% were unsure ($n = 72$), and four did not respond. Of the 1215 female respondents, 850 reported experiencing SHSA (70%), 300 did not (25%), 63 were unsure (5%), and 3 did not respond. Of the 259 male respondents, 43 reported experiencing SHSA (17%), 206 did not (80%), and 9 were unsure (3%). Of the 28 who identified as transgender, non-binary, or gender-fluid, 17 reported experiencing SHSA (61%), while 11 did not (39%). Gender differences in the proportion and type of SHSA incidents were explored (see Table 1).

The data were submitted to a mixed ANOVA to examine whether there were significant differences in the type of incidents experienced by gender. In support of the hypothesis, the main effect for gender was significant; $F(2, 1498) = 91.142$, $p < 0.0001$, $\eta^2 = 0.108$. Effect size statistics quantify the degree to which the findings can be viewed as having "practical significance" (Kirk 1996). Following Vacha-Haase and Thompson (2004) and Tomczak and Tomczak (2014), we chose $\eta^2$ as the effect size measure of choice. It measures the proportion of the total variance explained by gender after accounting for variance explained by other variables in the model. In this context, $\eta^2 = 0.108$ indicates that gender accounted for approximately 11% of the variance in reported SHSA incidents and can be considered a medium to large effect size. Bonferroni-adjusted post hoc tests indicated that female and non-binary, transgender, and gender-fluid respondents reported significantly more types of SHSA incidents than male respondents ($p < 0.001$).

**Table 1.** Gender differences in proportion and type of SHSA incidents.

| Type of SHSA Incident | Male (n = 259) | | Female (n = 1215) | | Transgender, Non-Binary, Gender-Fluid (n = 28) | | Total (n = 1502) | |
|---|---|---|---|---|---|---|---|---|
| | *M* | *SD* | *M* | *SD* | *M* | *SD* | *M* | *SD* |
| Catcalling | 0.278 | 0.449 | 0.657 | 0.475 | 0.429 | 0.504 | 0.587 [c] | 0.492 |
| Unwanted Verbal Sexual Advances | 0.104 | 0.306 | 0.528 | 0.499 | 0.500 | 0.509 | 0.455 [d] | 0.498 |
| Unwanted Sexual Advances | 0.100 | 0.301 | 0.452 | 0.498 | 0.429 | 0.504 | 0.391 [d] | 0.488 |
| Unwanted Touching | 0.062 | 0.241 | 0.173 | 0.378 | 0.143 | 0.356 | 0.153 [e] | 0.360 |
| Spanking | 0.069 | 0.255 | 0.167 | 0.373 | 0.214 | 0.418 | 0.151 [e] | 0.358 |
| Flashing | 0.042 | 0.202 | 0.158 | 0.365 | 0.071 | 0.262 | 0.136 [f] | 0.343 |
| Forceable Kissing | 0.004 | 0.062 | 0.020 | 0.139 | 0.143 | 0.356 | 0.019 [f] | 0.138 |
| Rape or Attempted Rape | 0.000 | 0.000 | 0.014 | 0.118 | 0.071 | 0.262 | 0.013 [g] | 0.112 |
| Forced Performance of Sexual Acts | 0.000 | 0.000 | 0.009 | 0.095 | 0.036 | 0.189 | 0.008 [g] | 0.089 |
| Total | 0.660 [a] | 1.198 | 2.178 [b] | 1.705 | 2.036 [b] | 2.333 | 1.913 | 1.739 |

Note. SHSA = sexual harassment and/or assault. SHSA incident scored either 0 (respondent reported no incidents) or 1 (respondent reported at least one such incident). [a,b] = Gender main effects with different superscripts differ beyond ($p < 0.05$) by a post hoc Bonferroni procedure. [c,d,e,f,g] = SHSA incident types sharing the same mean do not differ from each other; those with different superscripts differ from each other via the Scheffé procedure ($p < 0.05$).

Because the data violated sphericity, $W = 0.017$, $c^2(35) = 6079.68$, $p < 0.0001$, a necessary requirement for within-subject designs, Huynh–Feldt corrected tests were calculated and assessed for significance and effect size. These adjust for the sphericity violation and produce correct significance and effect size results. The main effect for the type of SHSA incident was also significant $F(5.155, 7727.799) = 61.271$, $p < 0.0001$, $\eta^2 = 0.039$. Within-subject and mixed-design effect sizes often present a challenge for correct interpretation (Morris and DeShon 2002), as different authors offer different interpretations of the meaning of a within-subject effect size itself. That said, a conservative view would be that the type of SHSA incident accounted for approximately 3.9% of the variance in reported SHSA incidents net all other variables in the design. While 3.9% may be seen as relatively trivial, it falls midway between what Cohen (1988) called a small and medium effect size. As such, it likely represents a true source of variation that indicates that some types of SHSA incidents were reported by more respondents than others and that the magnitude of these differences matters.

Differences due to SHSA incident type were assessed through a two-step process. First, the incident types were placed in descending order based on the magnitude of their total means. Second, a series of eight planned comparisons assessed differences between each adjacent pair of incident types (e.g., catcalling vs. unwanted sexual advances; unwanted sexual advances vs. unwanted verbal sexual advances). This procedure mirrors a post hoc Scheffé test to determine the number of separate sets of incident types. These sets all differ significantly from each other ($p < 0.05$), while the incident types within each set do not differ from each other. This process yielded five homogenous sets of incident type: Set I (catcalling) > Set II (unwanted verbal sexual advances, unwanted sexual advances) > Set III (unwanted touching, spanking) > Set IV (flashing, forcible kissing) > Set V (rape or attempted rape, forced performance of sexual acts).

The interaction effect was also significant, suggesting that differences among the three gender groups were not consistent across the nine SHSA incident types; $F(10.311, 7727.799) = 32.197$, $p < 0.0001$, $\eta^2 = 0.041$. As displayed in Figure 1, it appears that gender differences were relatively large for catcalling, unwanted verbal sexual advances, and unwanted sexual advances, while relatively smaller for the rest of the SHSA incident types. Examining the simple main effects tends to confirm this suspicion. The gendered $d_{MAX}$ values across the SHSA incident type vary substantially, from a minimum of 0.036 for the forced performance of sexual acts to a maximum of 0.424 for unwanted verbal sexual advances.

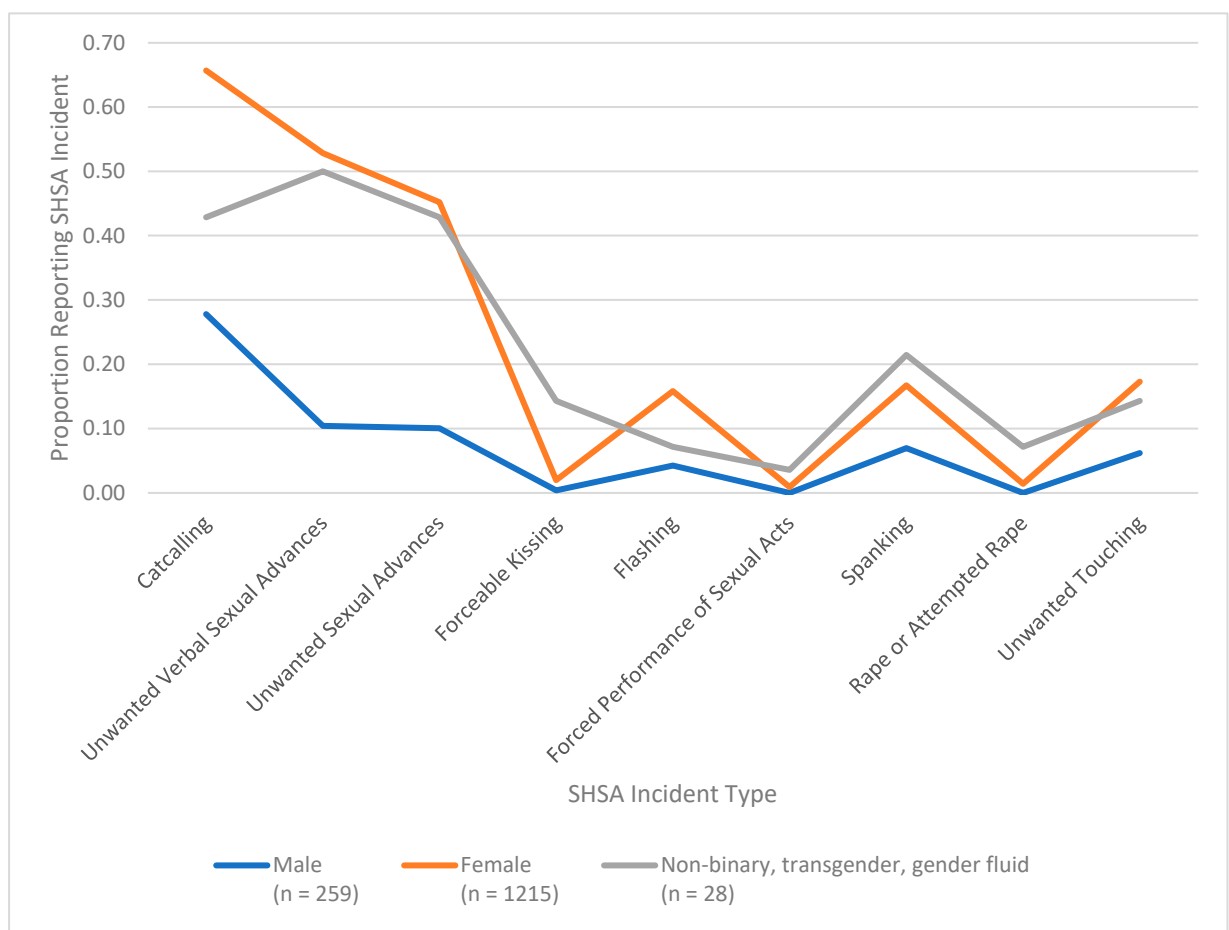

**Figure 1.** Proportion of reports of different types of SHSA incidents. Note: SHSA = sexual harassment and/or assault.

*3.1. Perceptions of Safety*

Comparing those identified as extreme responders (outliers) with those who were deemed normative responders is instructive because large differences would be expected between these two groups of respondents in relation to their perceptions of safety and the extent to which they modified their running behavior due to SHSA. The two groups differed substantially in terms of their perceptions of safety; $\chi^2(2, n = 1495) = 36.714, p < 0.0001$ (see Table 2). Only one-third of the outlier group (30.7%) reported feeling safe compared to two-thirds of the normative responders (63.9%). In essence, twice as many normative responders as extreme responders reported feeling safe in their running communities.

**Table 2.** Safety perceptions of the extreme and normal responder groups.

| | | Normative Responders ($n$ = 1420) | Extreme Responders ($n$ = 75) |
|---|---|---|---|
| **Question** | **Response** | **Frequency, $n$ (%)** | **Frequency, $n$ (%)** |
| Do you feel safe in your running community? | Yes | 907 (63.9) | 23 (30.7) |
| | Sometimes | 487 (34.3) | 47 (62.7) |
| | No | 26 (1.8) | 5 (6.7) |

Note. Missing responses = 7.

*3.2. Changes in Running Behavior*

Respondents were asked, "How did your experience with sexual harassment or sexual assault while engaged in a running-related activity change your relationship with the

sport?" There were substantial and significant differences between the normative and extreme responder groups ($\chi^2$(3, *n* = 1037) = 8.533, *p* < 0.04). Among the normative responders, approximately 61% indicated that it changed how they run and interact during running-related activities, while among the extreme responders, the number was even higher—69.3%. Over one-third of the normative responders (34.1%) and about a fifth of the extreme responders (22.7%) indicated that experiencing SHSA did not change their relationship with running.

### 3.3. Gender Differences in Perceived Safety and Changes in Running Behavior

Gender was also significantly related to perceived safety and the extent to which SHSA changed their running behavior. Over ninety percent (92.6%) of the male respondents reported feeling safe—a much larger percentage than either the female (55.9%) or transgender, non-binary, or gender-fluid respondents (57.1%). The differences were highly significant ($\chi^2$(4, *n* = 1496) = 122.825, *p* < 0.0001). An even larger and statistically significant difference was noted in respondents' responses to "How did your experience with sexual harassment or sexual assault while engaged in a running-related activity change your relationship with the sport?" ($\chi^2$(6, *n* = 1112) = 83.290, *p* < 0.0001). Approximately 74% of the male respondents indicated they did not change their running behavior after experiencing SHSA. In contrast, 30% of the female runners and 26% of the non-binary, transgender, or gender-fluid respondents reported not changing their running behavior in response to experiencing SHSA. These findings indicate that female and gender-diverse runners were more likely than male runners to have changed their running behavior and their overall relationship with running after SHSA.

### 3.4. Qualitative Analyses

Open-ended responses concerning feelings of safety within the running community were examined by two trained research assistants. Raters agreed 67.1% of the time (193 + 163 + 359)/1065. When corrected for chance agreement, the percentage drops to 59.4%, the value of weighted kappa (κ). This is statistically significant (κ = 0.594, *z* = 24.41, *p* < 0.001; 95% CI [0.555, 0.632]), representing fair to good agreement for low-stakes and exploratory research (Fleiss et al. 2003). For those who reported feeling unsafe in their running community, three common themes emerged: (1) the objectification of women, (2) the culture of acceptance, and (3) the necessity of changing running behavior.

The first theme that emerged from the qualitative open-ended responses was the objectification of women. Those who felt unsafe described ways in which women were often "objectified" and "preyed on" in their running communities, experiencing frequent catcalls, kissing, sexual comments, and unwanted touching. Women who ran faster and female race directors were more often targeted. Qualitative responses from six male respondents revealed that they acknowledged their societal and physical privilege, particularly those who identified as White males. One stated, "I think the simple reason why I feel more or less safe is the circumstance that I am male." Another stated, "I acknowledge that being a white male has given me the privilege not to have to deal with these unfortunate situations." Some stated that objectification has changed their relationship with running. A 30-year-old woman stated, "I've been sexually harassed, and touched inappropriately multiple times, catcalled, and belittled. I hear disparaging comments about women and gender-nonconforming folks from men when they are gathered around chatting. It makes me less inclined to participate."

A second theme that emerged was a culture of acceptance of SHSA. Those who felt unsafe described ways that SHSA has been "normalized" in the trail and ultrarunning community and at race events. Female survivors of SHSA were disheartened after they were "not taken seriously" when divulging to both men and women about the frequent objectification of women. When disclosing that certain men and race directors were persistent predators, they stated that they were angry because "everyone knows, yet there are no repercussions." One respondent said, "I've been conditioned to accept this behavior

from an early age . . . and thus experience it as 'normal.'" Another runner stated, "When I talk to other people in the running community about harassment or a culture of misogyny, many people react by either brushing it off or telling me to call the police—neither of which is a helpful response."

Men and gender-diverse runners also commented on this culture of acceptance of SHSA across trail and ultrarunning communities. A male runner stated, "Very often I have been catcalled while running in various communities around the U.S., especially when running shirtless." Another man stated, "My butt was grabbed twice after finishing a 5K. I can defend myself if needed, but it is uncomfortable when this happens. As a man, I am encouraged to suck it up or play it off as if it was not a big deal." A transgender runner stated, "I get verbally harassed regularly. When something happens, I get freaked out for a while, until I get over it and it fades from immediate memory." This culture of acceptance has contributed to runners feeling unsafe. This is exemplified by a 31-year-old runner who said, "I feel that many men and some women do not take this issue seriously, and it sometimes makes me feel alone and unsafe."

Many runners who reported SHSA felt they had to change their running behavior. Qualitative responses revealed that those who felt unsafe stated that they now consider when and where they run, whom they run with, and what they wear. Some runners who experienced SHSA indicated that it forced them to change their running community. After being raped in a running group, one runner said they felt safer running alone than in their running community. Some runners have made drastic changes to their running behavior, reflected by a female runner who said, "I won't run certain places when it's dark. I tell my partner where I am running and for how long. I keep my keys handy to use as a weapon. I change routes if there is a man coming towards me. I only listen to anything at low volume with one headphone." A transgender runner stated, "While running I've been stalked, targeted, and almost killed. At times I feel safe and other times I don't. I have had to change when and where I run and take many precautions."

Despite having experienced SHSA, many trail and ultrarunners' statements exhibit signs of resiliency. This is reflected by a runner who said, "I felt scared, threatened, and objectified, and I have spent a long time working through these incidents, but running is too important to me. I will not be dissuaded by men, stealing the thing I love the most." Some runners stated they were empowered to continue running after SHSA due to the encouragement of their running community. After experiencing catcalls, spanking, and unwanted touching, one runner said, "The majority of the people I train with every day are female, and they encourage me and empower me." Others stated, "When an incident occurs, the person gives our running group a head's up by posting the incident on our Facebook group page. I feel like our running groups all look out for each other;" and "I feel that the group around me is open to conversations and changing actions." A female Latina runner who experienced numerous catcalls, unwanted touching, flashing, and unwanted sexual advances stated, "There are lots of strong women in my community and we look out for each other, most importantly BELIEVE each other, and BELIEVE IN each other." These qualitative responses suggest that runners who are more resilient after having experienced SHSA have strong supportive relationships within their trail and ultrarunning community who will take action to protect one another and create change.

## 4. Discussion

Findings from the present study are consistent with SHSA studies across diverse sports and athletic activities (Hamilton 2017; Lieu and Rennison 2018). Sixty-one percent of respondents reported one or more incidents of SHSA while running. In support of predictions, female runners experienced a greater number of incidents of SHSA than those who identified as male, transgender, non-binary, or gender-fluid. In addition, transgender, non-binary, and gender-fluid runners reported higher rates of SHSA than male runners. Both quantitative and qualitative findings revealed that people of gender-minoritized groups encountered more SHSA in trail and ultrarunning communities, which mirrors

statistics across the nation. These results are congruent with national SHSA statistics verifying that one in three women, one in six men, and one in two transgender, non-binary, or gender-fluid individuals are sexually harassed and/or assaulted in their lifetime (James et al. 2016; Smith et al. 2018).

### 4.1. Experiences of SHSA and Gender Differences

Due to the lack of empirical studies of SHSA in the trail and ultrarunning communities, the present study aimed to explore the types and frequencies of SHSA in relation to gender differences. Results revealed five clusters of types of incidents, including (1) catcalling, (2) unwanted verbal sexual advances and unwanted sexual advances, (3) unwanted touching and spanking, (4) flashing and forcible kissing, and (5) forced performance of sexual acts, attempted rape, or rape. The most frequent types of reported incidents among runners were catcalls, unwanted physical and verbal sexual advances, and unwanted touching and spanking. Gender differences were significant, indicating that female runners and gender-diverse runners reported experiencing more catcalls and unwanted verbal and physical sexual advances than male runners. The findings support other studies indicating that women and gender-diverse athletes are more vulnerable to various types of SHSA while engaging in sports (Taha-Thomure et al. 2022; Vertommen et al. 2016). This highlights the larger scope of the problem in terms of the inequities and vulnerable populations. Further research is needed in trail and ultrarunning communities, examining when and where these incidents of SHSA occur and whether perpetrators are strangers or acquaintances.

### 4.2. Two Distinct Groups

The results revealed two distinct groups that varied in terms of frequency and types of SHSA incidents. While some reported relatively few SHSA incidents, 75 respondents experienced exceedingly frequent catcalls, spanking, flashing, unwanted verbal advances, stalking, and sexual assault while engaged in a running-related activity. This group reported between 1000 and over 100,000 incidents, suggesting that those who experience traumatic events have unique memories of their experiences. Otgaar et al. (2021) provide explanations for why individuals may not recollect traumatic experiences accurately, suggesting that some repress the memory to forget, and some do not want to talk about their experience, so it is not reinforced. Others may have initially thought the event was not traumatic but later reinterpret it as abusive. When attempting to recall traumatic events retrospectively, it may not be easy to retrieve accurate memories, or, alternatively, false autobiographical memories may develop (Brewin and Andrews 2016; Calado et al. 2020; Otgaar et al. 2021).

When trauma is experienced repeatedly over time, the perception and meaning of the experience may change, or memory amplification could occur (Southwick et al. 1997; van Giezen et al. 2005). In a study of veterans one month after returning from Desert Storm, Southwick et al. (1997) found that 70% changed their responses from initially reporting that certain events did not occur, and then two years later reported that they did recall the traumatic events. In a study of sexual assault victims, Nahleen et al. (2021) found that memory amplification did not occur, but they suggest that the accumulation of post-event information, media reports, and symptoms of distress may increase amplification errors as traumatic events are reappraised. In the present study, with so many participants reporting 1000 to over 100,000 incidents of SHSA while running, further research is needed to better understand the retrospective memories of survivors of SHSA. It is possible that those who reported an unfathomable number of incidents of SHSA while running (e.g., 100,000 unique incidents) may have reached a tipping point. That is, they have experienced SHSA so often that they put the highest number that could accurately represent their experience.

### 4.3. Perceptions of Safety

Of this outlier group who reported exceedingly frequent catcalls, spanking, flashing, unwanted verbal advances, stalking, and sexual assault, over 69% stated that they felt significantly less safe in their running communities compared to 36% of those who reported

few SHSA incidents. Subjective interpretations of experiences of SHSA may not be any less important than the number of reported incidents when treating post-traumatic stress, depression, anxiety, and intergenerational trauma. Rintaugu et al. (2014) found that having multiple incidents and various types of sexual harassment occur (e.g., sexually offensive looks and unwanted comments on attractiveness) caused physical ailments, including headaches, fatigue, and insomnia. In a qualitative study, 25 elite athletes described feeling disgust, fear, irritation, and anger after incidents of SHSA (Fasting et al. 2007). Whether it is an accurate number of reported incidents or one that best represents runners' subjective experiences, runners who feel unsafe due to SHSA are likely to experience negative repercussions for their physical and psychological well-being.

*4.4. Changes in Running Behavior*

Qualitative findings revealed that those who reported feeling unsafe in their running community described the objectification of women, a culture of acceptance, and the necessity to change their running behavior. This supported quantitative findings indicating that between 61% and 69% of those who reported SHSA while engaged in a running activity changed their running behavior. In congruence with Lieu and Rennison's (2018) findings, survivors of SHSA made drastic changes to their running routine, choosing not to run at certain times, holding their keys as a weapon, changing routes, and leaving their running community.

*4.5. Changing the Culture of Sports*

Often, victims are blamed or assumed to be at fault for not taking safety precautions or not reporting it in a timely manner. Some may question why a person would allow 1000 or 100,000 incidents of SHSA to occur. It is important to listen to and respect all runners' experiences and accounts of SHSA to help reduce the stigma around reporting SHSA and supporting mental health. With so many in the running community affected by SHSA, it is important to listen to survivors' stories to understand the way they develop meaning and define their experiences of SHSA. The results indicated that many of these runners are resilient and still feel empowered by their running communities. The present study supports ongoing efforts to provide a safe space for runners to share their experiences of SHSA to assess the prevalence of violence and gender inequities in running communities around the world. It is imperative that running communities, especially race directors, athletes, and community leaders, take action to make a change.

Gender disparities in incidents of SHSA for runners have raised heightened concern, but they provide an opportunity to discuss strategies for creating equitable and safe communities for trail and ultrarunners. To cultivate a safe space for runners and to change the culture of the running and race communities, we recommend multiple paths to fostering awareness and preventing SHSA. Runners and race directors should be encouraged to participate in implicit bias and diversity, equity, and inclusion (DEI) trainings to address the increased risk of harassment, violence, and inequities that minoritized and gender-diverse runners frequently encounter. It is important to create awareness of SHSA through increased communication and networking and to hold all athletes accountable while participating in running-related activities and events. One way to establish accountability standards is to develop a code of conduct, which includes a zero-tolerance policy for SHSA. Codes of conduct could mirror those used by businesses to reflect state or federal laws, which protect employees from SHSA by anyone regardless of sex, gender, or sexual orientation. At the October 2021 U.S. Trail Running Conference, we presented results from the current study and suggested mandating a code of conduct. Shortly thereafter, the race sponsor Salomon began developing required policy changes for their sponsored races and agreed to sponsor DEI workshops (T. Chiplin and J. Sanborn, personal communication, 22 January 2022). Similar codes of conduct could be established by race directors for all sponsored events and activities.

## 5. Limitations

The present study is the first empirical study to assess SHSA within the trail and ultrarunning community, so there are several limitations to be addressed. First, the study was administered online with a self-report measure. Due to the sensitive nature of this study, participants may not have answered honestly and provided socially desirable responses, which may lead to inaccurate self-reports.

Having unequal gender subgroups and a lack of a balanced sample also limits the generalizability of the findings. Due to the small subgroup identifying as transgender, non-binary, or gender-fluid, for purposes of analysis, these individuals were combined into one gender subcategory to protect anonymity. This was not ideal, but it was a way to protect the confidentiality and anonymity of our respondents.

*Suggestions for Future Research*

Two-Spirit was inadvertently left out as a gender category option. Two-Spirit is a term used within some Indigenous cultures in North America to describe individuals who embody both masculine and feminine qualities or spirits, which constitute third and fourth gender categories within Native American and Indigenous communities (Indian Health Services n.d.). There are disproportionately higher rates of the abduction and homicide of Indigenous girls and women due to a combination of systemic racism, violence, and inadequate responses from law enforcement and policymakers, prompting a call for the running community to advocate for justice and action to address the root causes and support Indigenous communities (Daniel n.d.). Future studies need to prioritize diversity, equity, and inclusion by recruiting more representative samples of marginalized gender and sexual minorities to identify populations that are more vulnerable to SHSA in trail and ultrarunning communities.

Further research is needed to identify the types of perpetrators (e.g., acquaintance, stranger, etc.), the locations SHSA most frequently occurs (e.g., on a secluded trail, at a race, etc.), and whether the number of incidents occurred in one setting or over time (e.g., did 10 catcalls occur in one setting or did 10 catcalls occur across various settings over time). Future research should build on this study to help trail and ultrarunning communities strategize ways of improving safety, cultivating empowerment, and reducing the rates of SHSA.

Sports are a source of empowerment for many girls and women (Heywood and Dworkin 2003). More research is needed to examine the protective factors sports may provide and the power of sports for expanding social capital and support networks to prevent SHSA (Raliance n.d.). Athletes, coaches, and families across running communities can be formidable mentors, role models, and spokespeople for social change. Races and running-related organizations can connect runners in the community, bringing people together for events as well as playing a unique convening role off the trails. By using these platforms to join people together around a shared vision, we can prevent and put an end to SHSA.

**Author Contributions:** The authors contributed to the paper as follows: Conceptualization, C.T.M. and C.S.T.; investigation, C.T.M., C.S.T., M.E. and J.M.; methodology, C.T.M. and C.S.T.; formal analysis, H.B., C.T.M., C.S.T., M.E. and J.M.; project administration, C.T.M.; draft manuscript preparation and writing—original draft, C.T.M., H.B., C.S.T., M.E. and J.M. All authors have read and agreed to the published version of the manuscript.

**Funding:** This research received no external funding.

**Institutional Review Board Statement:** The study was conducted in accordance with the Declaration of Helsinki. According to the Basic HHS Policy for Protection of Human Research Subjects and policies and procedures, this study was approved by the California State University Channel Islands Institutional Review Board (protocol code IO5565; approved 21 March 2021).

**Informed Consent Statement:** Informed consent was obtained from all subjects involved in the study.

**Data Availability Statement:** Data from the present study are available upon written request from the corresponding author. Due to the sensitive nature of this study, data are not publicly accessible due to ethical considerations.

**Acknowledgments:** We want to thank Terry Chiplin, facilitator of the US Trail Running Conference and Director of Active at Altitude (terry@ustrailrunningconference.com; activeataltitude.com), https://www.youtube.com/watch?v=5wGIvJw7zJ8&ab_channel=USTrailRunningConference and Jody Sanborn, Director of Prevention, Wyoming Coalition Against Domestic Violence and Sexual Assault. We appreciate the guidance provided by Callie Rennison (UC Denver) and Charlie Lieu, #SafeOutside while developing our survey. Thank you to Runner's World magazine and Taylor Dutch for supporting our data collection efforts. Finally, we are grateful for the valiant efforts of Raliance, a national collaborative to end sexual violence in one generation, joining forces with the National Alliance to End Sexual Violence (NAESV), the National Sexual Violence Resource Center (NSVRC), and California Coalition Against Sexual Assault (CALCASA).

**Conflicts of Interest:** The authors declare no conflict of interest.

## Note

[1]   Because the SHSA data severely violated both normality and homogeneity of variance requirements, they were not appropriate for *t*-tests. However, the *p*-values that emerged from the Mahalanobis analysis are appropriate for unequal variance *t*-tests (Rasch et al. 2007).

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
