# Peer review of "Sexual Harassment and Assault across Trail and Ultrarunning Communities: A Mixed-Method Study Exploring Gender Differences"

_socsci, doi:10.3390/socsci12060359_

Round 1
Reviewer 1 Report
Comment:
1. Note for p. 3
“Sexual Harassment: Physical and/or verbal harassment of a sexual nature, and unwelcome sexual advances, including catcalls, flashing, and requests for sexual favors.
Sexual Assault: Non-consensual sexual contact or behavior, including but not limited
to, forced kissing, fondling, unwanted touching, spanking, oral sex, and”
my comment: Please elaborate this paragraph of two concepts, so the reader may understand the exhalation of those two both categories, because in several countries, their rules still often considered these categories as the same and even exchanged those two. Please mention the references also.
2. This article is very interesting because apart from being an article that presents multidisciplinary studies, there are also not many studies on gender studies that describe gender construction in sports community and the culture. It’s also interesting to read the narration that displays various gender constructions. However, in the analysis section, I am still looking for explanations about the experiences of fellow runners who come from transgender or even non-binary groups while they’re facing SHSA.
Author Response
Response to Reviewer 1 Comments
Thank you very much for taking the time to review our manuscript and the helpful feedback you provided. We address each of your concerns in this revised version of our manuscript and provide responses to your feedback below.
Point 1: Sexual Harassment: Physical and/or verbal harassment of a sexual nature, and unwelcome sexual advances, including catcalls, flashing, and requests for sexual favors.
Sexual Assault: Non-consensual sexual contact or behavior, including but not limited
to, forced kissing, fondling, unwanted touching, spanking, oral sex, and" my comment: Please elaborate this paragraph of two concepts, so the reader may understand the exhalation of those two both categories, because in several countries, their rules still often considered these categories as the same and even exchanged those two. Please mention the references also.
Response 1: This is a very good point. We elaborated on our definitions in our Introduction under the section Prevalence of SHSA in Sports. We provide an additional review of an article by Lang et al. (2023) that examined 41 studies on gender-based violence in sports across 17 countries, describing the difficulty of comparing results across studies due to the wide range of definitions, research designs, methodologies, and samples used. Sexual harassment was most frequently used, and although sexual violence was the second most frequently used term, it encompassed sexual harassment and sexual abuse. The terms sexual abuse, sexual harm, sexual assault, emotional/psychological violence, physical violence, and homophobic violence were also frequently used across studies.
We provided a subheader Definitions at the end of our Introduction and added our citation (Kearl, 2018) for the definitions we used for sexual harassment and sexual assault.
Point 2: This article is very interesting because apart from being an article that presents multidisciplinary studies, there are also not many studies on gender studies that describe gender construction in sports community and the culture. It's also interesting to read the narration that displays various gender constructions. However, in the analysis section, I am still looking for explanations about the experiences of fellow runners who come from transgender or even nonbinary groups while they're facing SHSA.
Response 2: Thank you for this feedback. Great suggestion. We addressed this concern by describing our data analyses in more detail, as well as providing an additional Table and Figure to illustrate the impact of gender minoritized groups. In Table 1, we describe gender differences by frequency and types of SHSA incidents, showing the types of SHSA transgender, non-binary and gender fluid runners experience. In Figure 1, we show the profiles of different types of SHSA incidents by gender. We also added more qualitative responses for gender diverse runners.
Thank you again for your invaluable feedback and suggestions, which helped to improve the quality of our manuscript. We understand that with our new analysis and content, you may have additional feedback to provide. We welcome any other suggestions you might have, and look forward to your response.
Reviewer 2 Report
This manuscript is a very interesting examination of SHSA prevalence in the running community. It is very well-done, but I do have some comments/suggestions/questions.
Abstract
1. change 100k incidents to 100,000
Literature Review
2. Any previous research on different gender identities and their experiences of SHSA in other sports to add to the literature review?
3. Any specific SHSA research for trail/ultrarunners to include?
Methods
4. Please add a descriptive table
5. This is more of a descriptive paper since there are no multivariate analyses. Was there consideration of doing any multivariate analyses?
Discussion
6. First paragraph, 3rd sentence: would transgender, non-binary, and gender fluid individuals' experience more SHSA if the sample size was bigger? Discuss how this may have influenced your results with it being a smaller sample.
7. Page 8, 4th full paragraph--maybe add in something about diversity training for runners
Author Response
Response to Reviewer 2 Comments
Thank you very much for taking the time to review our manuscript and the helpful feedback you provided. We address each of your concerns through this revised version of our manuscript and provide responses to your feedback below.
Point 1: In Abstract change 100k incidents to 100,000
Response 1: Thank you for catching that. We edited it to 100,000.
Point 2: Literature Review Any previous research on different gender identities and their experiences of SHSA in other sports to add to the literature review? Any specific SHSA research for trail/ultrarunners to include?
Response 2: Thank you for this feedback. Unfortunately there were few studies examining experiences of SHSA for gender diverse runners. We found general statistics of SHSA experiences for gender minoritized individuals (James et al., 2016). Very few studies include gender-diverse athletes identifying as transgender, non-binary, gender fluid, two-spirit, etc. This is highly problematic since gender minoritized runners may be more vulnerable to SHSA while engaging in sports (Taha-Thomure et al., 2022; Vertommen et al., 2016). Taha-Thomure et al. (2022) conducted a case study of gender-based violence experience of a trangendered powerlifter. The other article Vertommen et al. (2016) was an excellent review examining interpersonal violence against children in sports, revealing that minoritized athletes including gender diverse and LGBTQ+ athletes experienced greater interpersonal violence in sports. We added this literature in our introduciton. We also discused the importance for future studies to prioritize research on gender minoritized runners due to the fact they may be more vulnerable to SHSA.
To date, there are no empirical studies examining the impact of SHSA on trail and ultrarunners. The present study aims to fill this gap.
Point 3: Discussion: First paragraph, 3rd sentence: would transgender, non-binary, and gender fluid individuals' experience more SHSA if the sample size was bigger? Discuss how this may have influenced your results with it being a smaller sample.
Response 3: Regardless of our small sample of gender diverse runners in our study, we found significant gender differences. Transgender, non-binary and gender fluid runners experienced a considerably higher number of incidents of SHSA than male runners. We added additional analyses, and included Table and Figure to illustrate the impact of gender minoritized groups. In Table 1, we describe gender differences by frequency and type of SHSA incidents, showing the types of SHSA transgender, non-binary and gender fluid runners experience. In Figure 1, we show the profiles of different types of SHSA incidents by gender. We also added more qualitative responses to highlight the experiences of gender diverse runners. We described the limitations and implications of having a small sample of gender diverse runners in the Discussion.
Point 3: Discussion: Page 8, 4th full paragraph--maybe add in something about diversity training for runners.
Response 4: We really appreciate this suggestion. In the Discussion section we added your suggestion for implicit bias and diversity training for runners and race directors, and mention where our work has contributed to making a policy change amongst race directors and sponsors.
Thank you again for your invaluable feedback and suggestions, which helped to improve the quality of our manuscript. We understand that with our new analysis and content, you may have additional feedback to provide. We welcome any other suggestions you might have, and look forward to your response.
Reviewer 3 Report
for detailed comments, please check the attachment.

Author Response
Response to Reviewer 3 Comments
Thank you very much for taking the time to review our manuscript and the helpful feedback you provided. We address each of your concerns in the revised version of our manuscript and provide responses to your feedback below.
Point 1: The study needs to significantly rewritten and streamlined into a cohesive unit. It is very confusing with multiple bits and pieces inserted which do not appear to link together in terms of the stated research goals of the study which indicates a prevalence study of SHSA in Trail running and ultra running communities, and an exploration of gendered differences in reported SHSA experiences, with an exploration of the questions identified on page 9 within the context of comparing those who report experiences of SHSA and those who do not …
Response 1: Thank you so much for your invaluable feedback. We revised our manuscript and streamlined our purpose and aims more clearly in our Introduction.
Point 2: The study seems to have got lost somewhere in the middle because of trying to justify the inclusion of the 75 unusual outlier data responders. The identification of the two groups is problematic to me. I would have expected two groups , one with zero reports of SHSA and one with reports of SHSA. But to identify two groups, both of which include reported experiences of SHSA, implies that the authors consider some { up to 2) reported experiences of SHSA to be equivalent to zero and therefore of no consequence to the victims. This is a very problematic position to take. The second issue with the grouping is the 75 outlier responders. The numbers are so vast as to raise questions of reliability of the data. I have further elaborated on this in the sections below. Essentially I think this data is too problematic to be included unless you have some access to follow up verification. It runs the risk of undermining the credibility of the whole study.
Response 2: This is a very good point. After careful consideration of whether to remove the outliers, we reanalyzed our data. Data were submitted to standard reliability and validity analysis, including testing distributional assumptions, the presence of extreme outliers, the probable effects of missing data and ways to minimize those effects. We describe our data screening process in a new section of our paper Data Screening and Analysis. Rather than eliminate the extreme responding participants, participant responses were dichotomized on all SHSA frequency data into two categories: (a) reporting no SHSA experiences, (b) reporting at least one SHSA experience. This transformed the SHSA responses into “No/Yes” (0, 1) data that fit most parametric analyses and almost all nonparametric analyses. In this way, data from all the respondents were included in the analysis. Missing data were deemed nonproblematic with fewer than 2% of the respondents having missing data.
Point 3: The study mentions equity but this is nowhere that I can see defined in terms of how this concept is incorporated into the structure, or discussion or proposed solution . On the contrary the reported experiences of SHSA of 17% of the male responders appear to be ignored and rendered invisible in the discussion. For example, in Page 15 a reference is made to national SHSA statistics { 1 in 3 women; and lin 2 transgender, non-binary, genderfluid individual, but no mention is made that the data for men is 1 in 4 {see NSVRC). we need to be careful of setting up false hierarchies of suffering . This is not equitable and we need to be careful not to silence or make invisible male victim's experiences of SHSA.
Response 3: You make a very important point here, and we will admit that our unconscious biases might have caused this oversight. We appreciate you pointing this out. We attempt to remedy this throughout our paper, providing a voice for the male, gender diverse, and ethnic/culturally minoritized runners who are survivors of SHSA.
We also point out under the Gender Disparities section that studies that only include female athletes perpetuate gender-based inequalities by not protecting male athletes who experience SHSA. Few studies include gender-diverse athletes identifying as transgender, non-binary, gender fluid, two-spirit, etc., which is highly problematic as they may be more vulnerable to SHSA while engaging in sports.
Point 4: I wonder if the title can be more concise and less journalistic and more specific to guide the reader as to what to expect .
Response 4: We changed our title to Sexual Harassment and Assault in the Trail and Ultrarunning Community: A Mixed Method Study Exploring Gender Differences. We hope this title is more acceptable.
Point 5: Need to be careful about language and specificity in reporting your own data and other studies. For example most studies are based on self- reports so we can only say the XXX% or number 'report' SHSA, rather than they experienced SHSA. Need to go through the document and be rigorous about this. Another example is the sentence "The chance of being harassed increased with performance level." This is not entirely correct and the language is not rigorous enough for a scholarly publications. It may be more correct to say "However, elite athletes ( 55%) were more likely to report SHSA than non-elite athletes (29%) indicating a possible increased risk of SHSA in elite level sport". In another example, in the Gender Disparities section, the first sentence reads "Gender disparities in incidents of SHSA are evident.. .. " It would be more correct to say "Gender disparities in reported incidents of SHSA are evident". Similarly in the second part of that sentence.
Grammatically you are using a lot of colon and Semi-colon sentence constructions, when two separate sentences would be the more correct presentation. Generally, we use colon and semi-colon construction when providing a list of things. Incorrectly using these constructions in the main text gives the impression that the authors have simply cut and pasted shorthand notes into the paper.
Response 5: Thank you for these suggestions. We worked on making sure that we are specific about reported statistics and findings. We also went through to examine our use of colons. It may be our writing style, and we were not aware of the way it reads.
Point 6: Paragraph 2 of the introduction should be paragraph 1 and include here the definition and participation rates in ultra running. Paragraph 1 should be paragraph 2. This will lead more logically into paragraph 3. IN the gender disparities section it seems the authors are setting scene to emphasise gender differences in reported prevalence.
Response 6: Much of our Introduction has been revamped and reorganized. Hopefully the manuscript flows better.
Point 7: The issue of perpetrators seems to be not part of the study so it is unclear why the text suddenly switches to perpetrators and likely causes. Additionally, the authors reference a 1986 study about "traditional Attitudes of masculinity contribute to aggressive sexual behaviors among emerging adult men".This is a very sweeping, over-generalization statement condemning all young adult men! There is no definition of what "traditional attitudes of masculinity" means.
Response 7: We appreciate you providing us with this constructive feedback. It was unintentional and not meant to overgeneralize or condemn young men (I have three boys so it is a huge concern of my own). We were trying to make a point about traditional expectations and beliefs about masculinity that are predictors of male dominant behaviors. In adolescence, traditional beliefs and expectations about masculinity are associated with cultural expectations for boys to initiate sex, be sexually aggressive with girls, and adhere to peer pressure, viewing sexual conquest as a validation for male adequacy (Zilbergeld, 1993). We edited this section to address this concern.
Point 8: The issue of the experiences of SHSA among marginalized gender participants should be introduced as a relevant element of the research in the previous Gender Disparities section.
Response 8: We added more literature on gender differences in SHSA across sports. A significant limitation of this research is that some of these studies do not include a representative sample or disaggregate data to examine important sub-categories, such as marginalized gender and sexual minorities. This is important for revealing inequities, exposing hidden trends, and identifying vulnerable populations to understand the scope of the problem entirely. Studies that only include female athletes perpetuate gender-based inequalities by not protecting male athletes who experience SHSA. Few studies include gender-diverse athletes identifying as transgender, non-binary, gender fluid, two-spirit, etc., which is highly problematic as they may be more vulnerable to SHSA while engaging in sports. We added the few articles we found on SHSA that include gender diverse individuals. The added information can be found primarily in the Gender Disparities section, as well as in the Limitations section.
Point 9: In the Trail and Ultra Running section, the authors need to conform to scholastic style, and reference appropriately Run Repeat is introduced with no explanation of that this organisation is and no reference for its study. Do not use journalistic terms like "staggering." The word "notwithstanding" in incorrectly used in the middle of the paragraph. The reference to the "gender gap" in participants not changing in 8 years is not referenced. What's the relevance of 8 years? Some information is missing here. The last sentence of this paragraph seems a non-sequitur and misplaced. It seems the purpose of this section is to explain what ultra-running is indicate the participation rates by gender? If yes, that this can be done in one or two sentences at the beginning of the study, ( see my comments above).
Response 9: We cited this reference appropriately and removed certain terms. We improved clarity by editing this section.
Point 10: What or who is Runner's Equity Alliance and what is the relationship to the study? Why is it introduced here? The current Focus group section is unclear. Runners Equity Alliance is introduced without explanation as to what this organisation is. No information of how many focus group were conducted , how many in each group, how the data was analyzed etc. Then 2 individual interviews seem to have also been conducted with representatives of #Safe outside and RALLIANCE. What happened to this data? Is this part of the study? If this was just a consultation to understand what they research are doing then you can just provide an acknowledgement at the end of the article rather than putting this unclear information in the text.
Response 10: The Runner's Equity Alliance was a group that initially started to band together a group of runners impacted by SHSA and to discuss the need with stakeholders to conduct this important study. They conducted a needs assessment in the trail and ultrarunning community to discuss the problems and the goals of the community to create prevention/intervention strategies to end SHSA. Unfortunately, the members went off to join other groups to promote their cause. The benefit and positive outcome of the group is that our research team continued on to conduct this study. We engaged the support and expertise of Raliance and #SafeOutside to create our survey questions. Callie Rennison and Charlie Lieu were compiling a large data set across a variety of sports. The benefit of collaborating with them is that we made our survey simple to follow their design so we could possibly later add our results to their data. The drawback is that our survey was too simple, and we did not ask key questions like who were the perpetrators (acquaintance, stranger, etc.), where the SHSA occurred (on a secluded trail or at a race), and whether the number of incidents occurred in one setting or over time (did they experience 10 catcalls at once by a group or 10 across different settings). We do acknowledge Raliance and #SafeOutside in our Acknowledgements.
Point 11: "One of the initial goals and objectives ... " What do you mean? Do you mean that the goals and objectives have since changed since this initial period"?
Three 3 aims of the study are presented here: to assess frequency of reported SHSA; to foster awareness of the prevalence of SHSA; and to develop proactive solutions.
The design and methodology should address how these aims are proposed to be met in the study in a logical and coherent manner . First, the design should be briefly explained. It seems the authors used a qualitative focus group approach to generate input for the quantitative part consisting of a questionnaire which also had a qualitative section of open-ended questions. There are 7 key questions identified in page 2 as constituting the main focus of the study. IN this descriptive results section, prior to reporting your analyses , you need to show the key descriptive data , including demographic data , the reported frequency , distribution of types of SHSA reported etc etc ( that is assuming you are going to report on all 7 research questions identified on page 2.) That way the reader has clear overview of the data set before you go into your analyses which should address your specific hypothesis and the research questions laid out on page 9.
Response 11: We outline the goals and aims more clearly in the Intro. We removed the seven key questions (used in a presentation at the Trail and Ultrarunning Conference). In a Research Design section, we explain the design and methodology of our study. We also created a Data Screening and Analysis section.
Point 12: I don't know what you mean when you refer to subjective vs objective experiences of SHSA. How can an experience be anything other than subjectively experienced??
Response 12: Objective experience is a question of whether they accurately recalled experiencing one incident, 1,000, or 100,000 incidents. Since it is a retrospective study, we are wondering if an accumulation of post-event information, media reports, and symptoms of distress increased memory amplification as they re-live and reappraise these traumatic experiences. It is possible that those who reported an unfathomable number of incidents of SHSA while running (e.g., 100,000 to 300,000 unique incidents) may have reached a tipping point. That is, they experienced SHSA so often that they put the highest number that could accurately represent their experiences. Our main point is that subjective interpretations of experiences of SHSA may not be any less important than the number of reported incidents when treating post-traumatic stress, depression, anxiety, and intergenerational trauma. Subjective interpretations of experiences are just as valid as the “actual” number of SHSA incidents they experienced.
Point 13: It seems this paragraph is dealing with the psychological impact of traumatizing SHSA experiences and you appear to be using this information to justify the 75 outlier responses. these 75 cases will raise questions as to credibility. Have you thought about deleting these 75 outlier cases and re run the data analysis without them . The risk with online public survey format is that credibility and reliability need to be closely considered.
Response 13: We decided not to delete these outlier cases and discuss this in Response 2 and within our manuscript.
Point 14: IN the section Perception of Safety, you are still referring to the two groups and their different perceptions of safety. But the following text seems to be reporting on the overall results reported on page 10 prior to separation into two groups. So the second sentence refers to the 850 female who did report SHSA s and you report their perception that they feels less safe, but there is no comparison , -less safe that who? I would have expected a comparison to contextualize the data with the group of women wo reported no SHSA.
Response 14: We rewrote this section with our new analyses, examining perceived safety, comparing those who reported experiencing a high number of SHSA incidents to those who experienced few.
Point 15: Table 2 seems not necessary - a simple statement is sufficient as the main information in not whether raters agree but rather the substantive qualitative themes.
Response 15: As per your suggestion, we removed Table 2.
Point 16: A separate section titled Emerging Themes seems not necessary. This information is more logically under the Perceptions of safety and changing running behaviour title . But again there is a mixing of issues, The first theme is objectification of women, but in the middle of that you are bringing up another theme of social/physical white male privilege? That should be a separate theme.
Response 16: Since we are highlighting this as a mixed method study, we changed this section with the subheader, Qualitative Analyses. We described our data analytic technique and the three themes that emerged from our qualitative analysis, providing examples and quotes from our respondents.
Point 17: Why did you only report national statistics on women and Trans, non-binary, gender fluid SHSA/ , but not men. Your data indicates that the reported SHSA among men - 17% is also similar to national data. see NSVRC). This is not an insignificant figure. And for me the message is a systemic lack of safety for anyone participating in ultra/trailrunning, which needs to be addressed.
Response 17: Thank you for catching this. We discuss the significance of this further, and the importance of disaggregating the data to understand the scope of the problem.
Point 18: Changing the Culture of Sport and creating safer spaces for runners.
You need to be more specific in this section and you need to expand it more with practical solutions because that is what you said your were going to do - . Go directly to your prescriptions for what needs to be done and address them specifically. The first two paragraphs here are vague. What is the prescription you are providing to inform the sector? The last sentence calls for action for change but does not specify what change based on your data?
Response 18: Per your suggestions, we specify ways in which the community can cultivate a safe space for runners and to change the culture of the running and race communities. We also address the importance of creating equitable communities for trail and ultrarunners. We suggest runners and race directors participate in implicit bias and diversity, equity, and inclusion (DEI) workshops to address the increased risk of harassment, violence, and inequities that minoritized runners and gender diverse runners frequently encounter. We suggest the importance of increased communication and networking, and to hold all athletes accountable while participating in running-related activities and events, establishing standards and developing a code of conduct, which includes a zero-tolerance policy for SHSA. Then we describe how after presenting at the Trail and Ultrarunning Conference, Salomon implemented our recommendations creating a mandatory code of conduct and sponsored DEI training.
Point 19: Limitation: I do not think it is a ,limitation that you did not formally assess the perpetrators. It seems that is a research question for another follow up study.
The limitations are the self-report nature of the study and the lack of generalizability because of the sampling methodology.
Response 19: Thank you for this suggestion. We rewrote this section addressing the limitations of the survey method and lack of generalizability and put suggestions for future research in another section.
Point 20: Protective Factors Empowering Runners
The first sentence in this section is not referenced or contextualised. It is followed by the concept that sport teaches young women positive body image and ability to make confident choices about relationships and sexuality. Linking these two sentences means we believe that women are victimized because they are not making good choices or by having poor body image. This is victim blaming. You really need to unpack this . Girls/Women, Trans, non-binary, gender fluid people, and boys/men are victimized because our society has a problem, not because they have a problem. Victimization occurs because of choices made by perpetrators not because of choices made by victims.
Response 20: Since this was beyond the scope of our study, we removed this reference from the study. However, we are extremely appreciative of your stance being an advocate for survivors of SHSA and calling for action to promote equity and inclusion of male victims as well as gender diverse groups who may not be recognized as vulnerable groups.
We thank you again for your detailed and constructive feedback. We attempted to address each concern outlined in your review. We understand that with our new analysis and content, you may have additional feedback to provide. We welcome any other suggestions you might have, and look forward to your response.
Round 2
Reviewer 3 Report
extensive revisions have been made and it now reads much more fluidly and coherently.